# Enhancing the Solubility of Curcumin Using a Solid Dispersion System with Hydroxypropyl-β-Cyclodextrin Prepared by Grinding, Freeze-Drying, and Common Solvent Evaporation Methods

**DOI:** 10.3390/pharmacy8040203

**Published:** 2020-11-02

**Authors:** Nguyen Ngoc Sao Mai, Riko Nakai, Yayoi Kawano, Takehisa Hanawa

**Affiliations:** Faculty of Pharmaceutical Sciences, Tokyo University of Science, 2641 Yamazaki, Noda, Chiba 278-8510, Japan; apricotd2003@gmail.com (N.N.S.M.); 3b16064@ed.tus.ac.jp (R.N.)

**Keywords:** solid dispersion(s), solubility, cyclodextrin, physicochemical properties

## Abstract

Cyclodextrins (CDs) and their derivatives significantly increase drug solubility by forming drug/CD complexes known as solid dispersions (SDs), which consist of an inclusion complex (IC), where the drug is entrapped within the CD cavity, and a non-IC. Here, the SDs of curcumin (CUR) and hydroxypropyl-β-cyclodextrin (HPβCD) were prepared using the grinding, freeze-drying (FD), and common solvent evaporation (CSE) methods and were physicochemically characterized using solubility, powder X-ray diffraction, Fourier transform infrared, differential scanning calorimetry, and dissolution studies. The second or higher order complex of CUR-HPβCD indicated the co-existence of ICs and non-ICs known as the SD system. When comparing the soluble drug amount with CUR crystals, the solubility of SDs was enhanced by up to 299-, 180-, and 489-fold, corresponding to the ground mixtures (GMs), freeze-drying mixtures (FDs), and common solvent evaporation mixtures (CSEs), respectively. The total transformation into the amorphous phase of CUR was observed in GMs and in CSE12, CSE14, and CSE18. The drug was well dispersed within HPβCD in GMs and CSEs, suggesting the formation of hydrogen bonds between CUR and HPβCD, whereas the dispersed behavior of FDs was similar to that of physical mixtures. In SDs, the melting temperature of CUR was in an increased order of CUR in 1:2 ICs, CUR in 1:1 ICs, and CUR crystals. The dissolution rate of CUR was positively improved as the amount of HPβCD in SDs increased. The SD system consisting of CUR and HPβCD significantly increased the drug solubility compared to ICs.

## 1. Introduction

Based on the solubility and permeability of active pharmaceutical ingredients (APIs) that regulate drug absorption, Amidon et al. divided drugs into the following four categories [1]: class I (high solubility–high permeability), class II (low solubility–high permeability), class III (high solubility–low permeability), and class IV (low solubility–low permeability). Approximately 70% of new pharmaceutical substances are classified as class II and class IV drugs, meaning they have poor aqueous solubility [2]. To address this, one of the basic goals of drug formulations is to improve the solubility of these two drug classes. Indeed, numerous strategies such as spray-drying [3], co-solvency [4,5], solid dispersion [6,7,8], complexation [9,10], and hydrotrophy [5] are used to increase drug absorption, resulting in higher bioavailability.

In particular, solid dispersion (SD) is one of the most promising strategies for enhancing the drug solubility owing to its ability to reduce the drug particle size [11], increase the drug wettability [12], develop porous systems [13], and reduce the drug crystallinity [14]. SDs refer to mixtures consisting of at least two components: an API and a hydrophilic carrier. In this system, the API is dispersed in a matrix of the hydrophilic carrier in a solid state. Various methods, such as hot-melt extrusion [15], spray-drying [16], freeze-drying (FD) [17], precipitation [18], solvent evaporation [19], and ball milling [20], have been used to prepare SDs.

From their first isolation in 1891, cyclodextrins (CDs) have been introduced as powerful substances to enhance drug solubility by forming host–guest inclusion complexes (ICs) [21,22,23,24,25,26,27,28]. CDs are cyclic oligosaccharides consisting of six, seven, or eight α-(1,4) linked glucopyranoside units, corresponding to α-, β-, and γ-CDs, respectively. They have toroid shapes with a larger and a smaller opening (Figure 1A). The CDs can increase the solubility of drugs that are entrapped within their hydrophobic interior because of their superior hydrophilic exterior when exposed to water. However, a drug/CD system not only consists of ICs but non-ICs as well [29,30]. In other words, the host–guest ICs are dispersed in the matrix of “empty” CDs as an SD system, which further enhances the solubility of ICs due to the hydrophilic exterior of CDs. Moreover, modified CDs using different functional groups express a superior possibility of improving drug solubility than conventional CDs [31,32].

In this study, curcumin (CUR) was selected as a model API, which is poorly soluble in water. CUR is a diarylheptanoid, which comprises two aromatic rings joined by a seven-carbon chain (Figure 1B). This chemical structure is responsible for its yellow color and low aqueous solubility (~28.9 ng/mL); thus, CUR is classified as a biopharmaceutical classification system class IV drug. An SD system consisting of CUR and a CD was prepared to simultaneously evaluate the ICs and non-ICs with an aim of increasing the CUR solubility. Akbik et al. proved that the ability of CDs to enhance CUR solubility gradually increased in the order of γ-CD, β-CD, methyl-β-CD, and hydroxypropyl-β-cyclodextrin (HPβCD) [33]. HPβCD is a derivative of β-cyclodextrin and its water solubility is over 500 mg/mL at room temperature [34]. Besides, the amorphous nature of HPβCD helps with the amorphization when dispersing drugs or drug/HPβCD ICs in the matrix of HPβCD in the SD system [35]. To the best of our knowledge, we considered an SD system consisting of CUR-HPβCD ICs and non-ICs instead of a conventional host–guest complexation [36,37,38]. The co-existence of ICs and non-ICs in SDs corresponded to the molar ratio of CUR and HPβCD, and to the methods of SD preparation (grinding, freeze-drying, and common solvent evaporation).

## 2. Materials and Methods

### 2.1. Materials

CUR was purchased from Tokyo Chemical Industry Co. Ltd. (Tokyo, Japan). HPβCD was purchased from Nihon Shokuhin Kako Co. Ltd. (Tokyo, Japan). MeOH was purchased from Kanto Chemical Co., Inc., (Tokyo, Japan). Ammonia (10% solution) was purchased from Wako Pure Chemical Industries, Co. Ltd. (Osaka, Japan). All reagents were of analytical grade.

### 2.2. Quantification of CUR

The CUR content was analyzed using ultraviolet (UV)–visible spectroscopy at 432 nm (Shimadzu 1800 UV–visible spectrophotometer, Shimadzu Co. Ltd., Kyoto, Japan). The analytical method was validated according to International Conference of Harmonization guidelines [39] in terms of specificity, linearity, accuracy, and precision. The standard CUR solution used for quantification was prepared in 50% *v*/*v* MeOH/distilled water (DW).

### 2.3. Phase Solubility Analysis

The phase solubility analysis was conducted using a method previously described by Higuchi and Connors [40]. Several concentrations of HPβCD (0, 5, 10, 15, and 20 mM) were dissolved in 10 mL DW in 20 mL L-tubes. CUR (20 mg) was added to the solution. Each tube was capped and shaken continuously for 72 h in a water bath at 30 ± 1 °C. Following equilibrium, these samples were filtered through a 0.45 µm membrane filter and assayed for the CUR content using UV–visible spectroscopy at 432 nm, as described in Section 2.2. The assay of each sample was carried out in triplicate.

### 2.4. Method of Preparation of SDs

#### 2.4.1. Grinding Method

The defined weights of CUR and HPβCD were mixed in a 30 mL glass sample tube with a vortex mixer for 60 sec to obtain the physical mixtures (PMs). Each PM (500 mg) was transferred to a 5 mL stainless steel jar on a mixer mill (MM400, Retsch, Haan, Germany) containing a ball (stainless steel, Φ7 mm). The jar was immersed in liquid nitrogen for 5 min and then the material was ground using the MM400 for 15 min at 30 Hz. The stroke immersion for 5 min in liquid nitrogen and the grinding was repeated twice, thus corresponding to 30 min of grinding time. Following this, the ground mixtures were passed through a 36-mesh sieve and stored in a desiccator until further evaluation.

#### 2.4.2. Freeze-Drying (FD) Method

The definite weights of HPβCD were dissolved in 20 mL DW with a small amount of 10% ammonia solution (pH = 10). Once CUR was added to the solution, the dispersions were stirred at 500 rpm for 15 min at 30 °C. Next, the dispersions were sonicated for 4 h at a temperature under 30 °C. The samples were then dried using a freeze-dryer (EYELA DFU-2110, Tokyo Rikakikai Co, Ltd., Tokyo, Japan) and passed through a 36-mesh sieve. Samples were stored in a desiccator until further evaluation.

#### 2.4.3. Common Solvent Evaporation (CSE) Method

The defined weights of HPβCD and CUR were dissolved in 50 mL MeOH. The mixtures were stirred at 30 °C, 400 rpm until a clear solution was obtained. The solvent was evaporated using a rotary vacuum evaporator (EYELA, Tokyo Rikakikai Co. Ltd., Tokyo, Japan) at 40 °C at a speed of 50 rpm. Then, the samples were placed in a vacuum dryer (vacuum sample drying oven HD-120, Ishii Laboratory Works Co. Ltd., Osaka, Japan) for 24 h to completely evaporate the solvent. The dried samples were passed through a 36-mesh sieve and stored in a desiccator until further evaluation.

### 2.5. Characteristic Evaluation Methods

#### 2.5.1. Solubility

Prepared samples containing 5 mg CUR were dispersed in 10 mL DW in 20 mL L-tubes. These tubes were capped and shaken continuously for 2 h in a water bath at 30 ± 1 °C until the solution attained equilibrium. Supersaturated solutions were filtered through a 0.45 µm membrane filter and further diluted with MeOH 50% *v*/*v* to obtain a suitable concentration within the calibration range. The content of CUR was analyzed using UV–visible spectroscopy at 432 nm, as described in Section 2.2. Each sample was analyzed in triplicate.

#### 2.5.2. Powder X-ray Diffraction (PXRD)

PXRD were performed using a diffractometer (RINT 2000, Rigaku Co. Ltd., Tokyo, Japan) using Cu Kα radiation, an X-ray tube voltage of 40 kV, and an X-ray tube current of 40 mA. The diffractograms were recorded in the 2θ range from 5° to 30° at a scan rate of 2°/min.

#### 2.5.3. Fourier Transform Infrared Spectroscopy (FTIR)

The attenuated total reflection method was used and FTIR spectra were recorded using a Fourier transform infrared spectrometer Frontier T-UATR (KRS5) (Perkin-Elmer Inc., CT, USA). The scanning range was set as 4000–400 cm^−1^ with an accumulation count of 16, a sample thickness of 1 mm, and a resolution of 1 cm^−1^. The obtained spectra were normalized by the standard normal variate (SNV) method [41].

#### 2.5.4. Differential Scanning Calorimetry (DSC)

DSC measurements were carried out using the DSC-60 Plus Differential Scanning Calorimeter (Shimadzu Co. Ltd., Kyoto, Japan), associated with the TA-60WS thermal analyzer (Shimadzu Co. Ltd., Kyoto, Japan) and the FC-60A flow controller (Shimadzu Co. Ltd., Kyoto, Japan). Samples were weighed (3~5 mg) and sealed in aluminum pans under N_2_ gas with a flow rate of 50 mL/min. An empty pan was used as the reference. The samples were scanned at 10 °C/min from 25–250 °C.

#### 2.5.5. Dissolution Studies

The dissolution profiles of CUR in the samples were evaluated using the dissolution apparatus as specified in the 17th Japanese Pharmacopoeia [42]. SDs containing 5 mg of CUR were accurately weighed and placed in a vessel containing 500 mL DW. The temperature was controlled at 37 ± 0.5 °C and the rotating speed of the paddle was set at 100 rpm throughout the studies. At defined time intervals (1, 3, 5, 10, 15, 30, 45, 60, 90, and 120 min), 5 mL aliquots were withdrawn, and replaced with an equal volume of fresh DW. The sample solutions were filtered through a 0.45 µm membrane filter and assayed for CUR content using UV–visible spectroscopy at 432 nm, as described in Section 2.2. Each sample was analyzed in triplicate.

#### 2.5.6. Dissolution Efficiency (DE)

Khan and Rhodes suggested DE as a suitable parameter for the evaluation of in vitro dissolution [43]. The term DE for a pharmaceutical dosage form is defined as the area under the dissolution curve up to a certain time *t*, expressed as a percentage of the area of the rectangle described by 100% dissolution at the same time. It can be calculated using the following equation:(1)$$\mathrm{DE}=\frac{\int_{0}^{t}y\times \mathrm{dt}}{y_{100}\times t}\times 100$$
where y is the drug percent dissolved at time *t*.

Although the CUR SD formation can increase the solubility of drug, the DE hardly attains 100%. As a result, we define the DE at a time *t* as a percentage of soluble drug at the same time. Therefore, the DE over 120 min of a dissolution study was calculated as a comparative parameter to evaluate the variation of drug kinetic solubility.

## 3. Results and Discussion

### 3.1. Quantification of CUR

At wavelength 432 nm, which is the *λ_max_* of CUR absorption, there was no interference absorption from HPβCD. The calibration curve of the method was linear (r^2^ = 0.9999) in the 0.1–10 µg/mL range. The recovery 99.26 ± 0.51% and the %RSD value < 2 indicated the high accuracy and precision of the method.

### 3.2. Phase Solubility Analysis

The phase solubility diagram of CUR and HPβCD is presented in Figure 2. The plot shows that the aqueous solubility of CUR increases with increasing HPβCD concentration up to 20 nM. According to Higuchi and Connors [40], this diagram is classified as A_P_ type, indicating the formation of second and/or higher order complexes with regard to HPβCD. However, the profile is not fitted to any equations which are used to calculate the stability constant of the complex. It has been suggested that A_P_-type profiles are strongly similar to phase solubility diagrams of lipophilic water-insoluble drugs in aqueous surfactant solutions [30].

Using ^1^H nuclear magnetic resonance (NMR) and 2D rotating frame overhause effect spectroscopy (ROESY), Jahed et al. reported that one or two aromatic rings of CUR entered the β-CD cavity [44] corresponding to 1:1 or 1:2 host–guest ICs (Figure 3A,B), respectively. Because of the existence of ICs and non-ICs in the CUR/HPβCD SD system, it is supposed that the SD consists of free CUR molecules, CUR in ICs, CUR in non-ICs, and “empty” HPβCD molecules. Thus, the components of CUR-HPβCD SD are observed as in Figure 3C. Moreover, the number of these components corresponded to the molar ratio of CUR-HPβCD and the preparation methods. For example, if there is an excess amount of HPβCD and dispersed force from the preparation method is large enough to form the ICs, the free CUR crystals and 1:1 ICs will disappear.

### 3.3. Solubility Study

The CUR solubility results are shown in Table 1. As the HPβCD molar was increased from 1 to 8 while the CUR molar was kept constant at 1, the CUR solubility ranged from 0.89–6.61, 0.27–5.23, and 2.64–14.16 µg/mL, corresponding to the grinding mixtures (GMs), freeze-drying mixtures (FDs), and common solvent evaporation mixtures (CSEs), respectively. An increase in the amount of HPβCD enhanced the solubility of CUR regardless of the preparation method. When comparing the soluble drug amount with CUR crystals, the solubility of SDs was enhanced by up to 299-, 180-, and 489-fold, corresponding to the GMs, FDs, and CSEs, respectively.

Li et al. reported that the CUR-HPβCD IC prepared by cosolvency-lyophilization had a drug loading capacity of 1:7 of the drug/CD molar ratio [38]. Thus, the aqueous solubility of CUR was enhanced to 15.2 mg/mL. Because two benzene groups of one molecule of CUR will be entrapped within two molecules of CD, corresponding to 1:2 of the CUR/CD molar ratio, Li et al. formed an SD system consisting of ICs and non-ICs but named it an IC only.

### 3.4. PXRD Patterns

Figure 4A presents the PXRD patterns of CUR crystals, HPβCD, and PMs of CUR-HPβCD with various molar ratios. CUR crystals showed prominent peaks at 2θ *=* 8.88°, 12.3°, 14.54°, 17.26°, 21.2°, 23.32°, and 24.72°, representing their crystalline nature, whereas HPβCD showed two broad peaks in the ranges of 10–15° and 15–20°, suggesting its amorphous nature. In PMs, the characteristic peaks of CUR are visible, but there are reductions in the intensity as the amount of HPβCD in the SDs increased.

Figure 4B shows halo patterns of CUR-HPβCD SDs prepared by the grinding method, indicating a total transformation from a crystalline form to an amorphous form. In Figure 4C, some prominent peaks of CUR were observed at 2θ = 8.92°, 12.34°, and 17.32°. However, peaks in the FD18 sample almost disappeared, thus demonstrating the amorphization of CUR. Figure 4D shows that in the PXRD patterns of the CSE11 sample, a peak at 2θ = 13.86° is observed, resulting in the crystalline form of CUR. However, in the CSE12, CSE14, and CSE18 samples, the prominent peaks of CUR are invisible, which suggests amorphization.

### 3.5. FTIR Spectroscopy

The prominent peaks observed in CUR are as follows: (1) 1602 cm^−1^ for the stretching vibration of the benzene ring skeleton; (5) 1506 cm^−1^ for the mixed (C=O) and (C=C) vibrations; and 1275 cm^−1^ for the methyl aryl ether (O-CH_3_) stretching vibrations [45]. Figure 5A shows the FTIR spectra of CUR crystals, HPβCD, and PMs of CUR-HPβCD with various molar ratios at a wavelength from 1000–2000 cm^−1^. As the amount of HPβCD in PMs increased, CUR peaks were obscured by HPβCD spectra.

New peaks could be observed at 1506 cm^−1^ in the case of GMs and CSEs (Figure 5B,D respectively), whereas FDs showed similar peaks as PMs (Figure 5C). It was supposed that CUR was well dispersed within HPβCD in SDs prepared by grinding and CSE methods and the dispersed behavior of FDs was similar to that of PMs. Additionally, the intensity of these new peaks positively decreased as the amount of HPβCD in mixtures increased. These new peaks were suggested to due to hydrogen bond formation between the oxygen of the CUR carbonyl group and the hydrogen of the HPβCD hydroxyl group.

### 3.6. DSC

The DSC curve for CUR crystals showed one endothermic peak at 188 °C, corresponding to the melting point of CUR. An endothermic peak at 222 °C was observed in the DSC curve of HPβCD, thus indicating its decomposition peak (Figure 6A). In thermograms of PMs, the melting peaks of CUR and decomposition peaks of HPβCD were detected, but they were shifted to lower temperatures. Because the mixtures of CUR and HPβCD are considered impure substances, this phenomenon is observed [46]. Furthermore, the shift range positively corresponds to the amount of HPβCD in PMs. On the other hand, the impurity of the 1:2 IC is considered to be lower than that of the 1:1 IC. Therefore, the endothermic peak of the 1:2 IC might be detected at a lower temperature than that of the 1:1 IC.

Figure 6B shows the DSC thermograms of GMs at various molar ratios. In the GM11 curve, it is supposed that the small and broad endothermic peak at 167.5 °C indicates the formation of ICs. In the case of GM12, there were two glass transitions at 163 and 188.9 °C, which correspond to the 1:2 and 1:1 complexes, respectively. In the GM14 curve, one glass transition was identified at 147 °C, which corresponds to the 1:2 complexes. However, in the GM18 graph, the glass transition is invisible because of a large amount of HPβCD, which may over-impose the signal of the transition peak.

Figure 6C,D reveal the DSC figures of FDs and CSEs, respectively. Because the melting point of a pure substance is higher and has a smaller range than the melting point of an impure substance [46], the melting temperature of CUR crystals is higher than that of 1:1 complexes and 1:2 complexes. Therefore, four endothermic peaks in CSE11 (Figure 6D) at 154.7, 171.7, 181.5, and 240.9 °C corresponded to the 1:2 complexes, 1:1 complexes, CUR crystals, and HPβCD, respectively.

### 3.7. Dissolution Study

The dissolution profiles of CUR crystals and SDs that were prepared by the grinding, FD, and CSE methods are depicted over a 120 min period (Figure 7). In general, SDs containing a higher amount of HPβCD released a higher concentration of CUR and, at the same molar ratios, CSEs had a higher dissolution rate than GMs and FDs.

According to the dissolution profiles of PMs (Figure 7A), CUR is gradually released, and its concentration tends to increase over the 120 min of this study. However, GMs exhibit different behavior as the molar ratios of components differ (Figure 7B). With molar ratios of 1:1 and 1:2, the released CUR peaks at 3 min, and then slightly decreases before plateauing over the remaining time (5–120 min). Alternatively, in GM14 and GM18, the soluble CUR rapidly peaks after one minute, decreases from 3–30 min, and plateaus over the remaining time (45–120 min). It is observed that there is a similarity in the dissolution behavior of GM14 and GM18.

The dissolution profiles of the FDs (Figure 7C) show the fast release of CUR concentration over 1-5 min, a slight variation over 10–60 min and stable release over the remaining time (90–120 min). For CSEs, CUR was released very quickly during the first minute and continued to increase its concentration up to 15 min (Figure 7D). However, the concentration of released CUR slightly decreases over the remaining time (30–120 min).

### 3.8. DE

The DEs (%) over 120 min of CUR crystals and SDs prepared using the grinding, FD, and CSE methods are displayed in Table 1. As the HPβCD molar was increased from 1 to 8 while the CUR molar was kept constant at 1, the DE increased from 0.66 to 4.31% in GMs, from 1.48 to 4.69% in FDs, and from 2.82 to 52.21% in CSEs. The similarity in the dissolution profile of GM14 and GM18 was due to their DEs of 4.02 and 4.69, respectively. The CSE18 sample gave the best DE of 52.21%, 145-fold better than DE of CUR crystals (0.36%).

Because the CSE method was considered to form entirely ICs with the molar ratio of CUR:HPβCD = 1:2, the SD of CSE12 consisted of 1:2 ICs only. In other words, there was no CUR crystal, “empty” HPβCD, and 1:1 IC in the CSE12 whose DE was 10%. When the amount of HPβCD increased (in CSE14 and CSE18), the SD system was obtained and the DEs of CSE14 and CSE18 were 31.27 and 52.21%, respectively. This indicated that the dispersion of ICs in the matrix of HPβCD significantly increased the drug solubility compared to ICs.

## 4. Conclusions

In this study, the SDs of CUR and HPβCD were prepared using the grinding, FD, and CSE methods and were physicochemically characterized using solubility, PXRD, FTIR, DSC, and dissolution studies. The second or higher order complex of CUR-HPβCD indicated the co-existence of ICs and non-ICs, known as the SD system. When comparing the soluble drug amount with CUR crystals, the solubility of SDs was enhanced by up to 299-, 180-, and 489-fold, corresponding to the GMs, FDs, and CSEs, respectively. The total transformation into the amorphous phase of CUR was observed in GMs, CSE12, CSE14, and CSE18. The drug was well dispersed within HPβCD in GMs and CSEs, suggesting the formation of hydrogen bonds between CUR and HPβCD, whereas the dispersed behavior of FDs was similar to that of PMs. In SDs, the melting temperature of CUR was in an increased order of CUR in 1:2 ICs, CUR in 1:1 ICs, and CUR crystals. The dissolution rate of CUR was positively improved as the amount of HPβCD in SDs increased. The SD system consisting of CUR and HPβCD significantly increased the drug solubility compared to ICs.

## Figures and Tables

**Figure 1 pharmacy-08-00203-f001:**
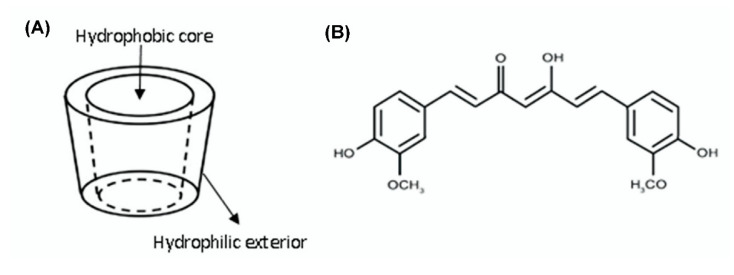
(**A**) Toroidal shape of cyclodextrins; (**B**) chemical structure of curcumin.

**Figure 2 pharmacy-08-00203-f002:**
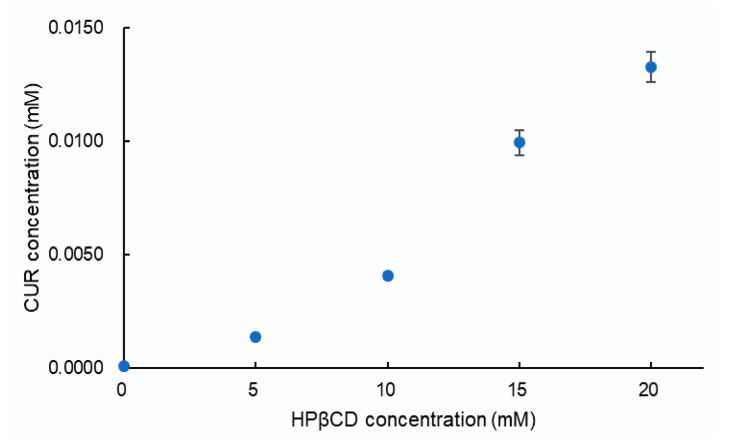
Phase solubility analysis.

**Figure 3 pharmacy-08-00203-f003:**
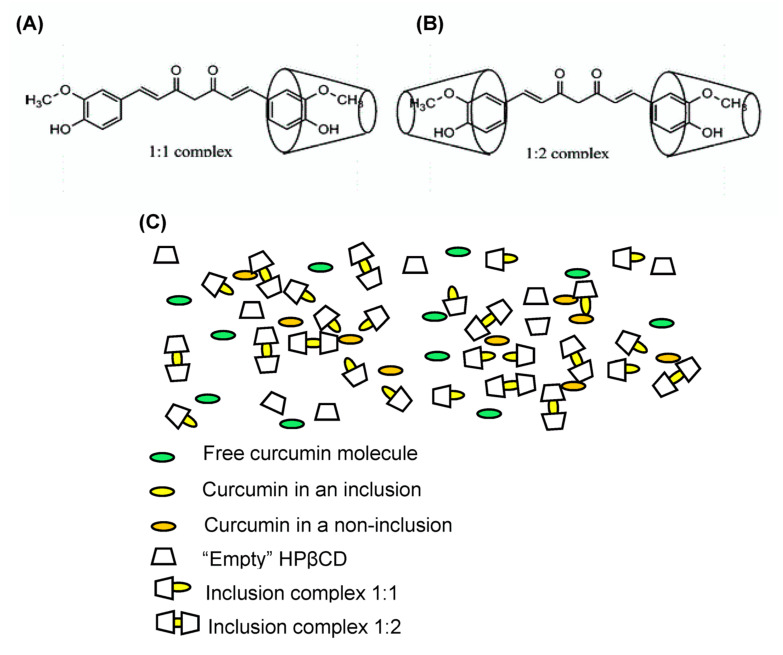
Schemes of: (**A**) 1:1 curcumin (CUR): hydroxypropyl-β-cyclodextrin (HPβCD) complex; (**B**) 1:2 CUR:HPβCD complex; and (**C**) components of CUR:HPβCD solid dispersions.

**Figure 4 pharmacy-08-00203-f004:**
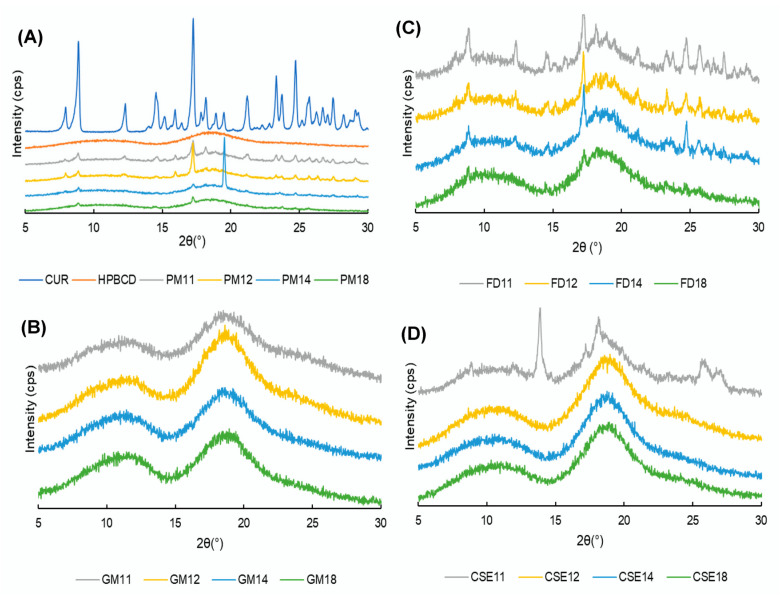
Powder X-ray diffraction patterns of: (**A**) curcumin (CUR), hydroxypropyl β cyclodextrin (HPβCD), and physical mixtures (PMs); (**B**) ground mixtures (GMs); (**C**) freeze-drying mixtures (FDs); and (**D**) common solvent evaporation mixtures (CSEs).

**Figure 5 pharmacy-08-00203-f005:**
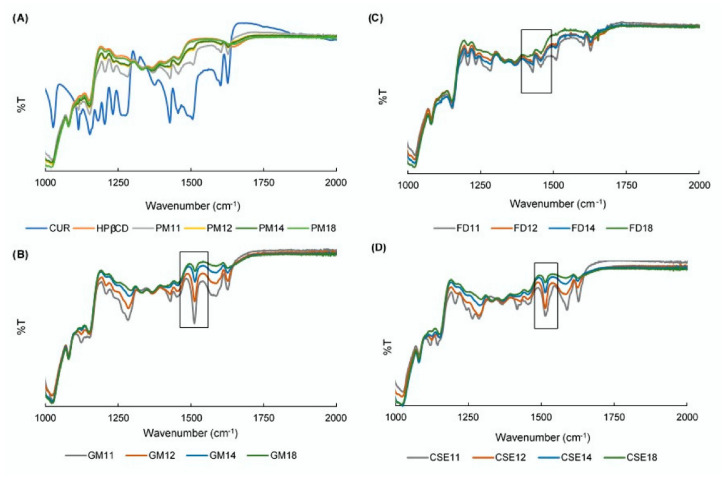
Fourier transform infrared spectra of: (**A**) curcumin (CUR), hydroxypropyl β cyclodextrin (HPβCD), and physical mixtures (PMs); (**B**) ground mixtures (GMs); (**C**) freeze-drying mixtures (FDs); and (**D**) common solvent evaporation mixtures (CSEs).

**Figure 6 pharmacy-08-00203-f006:**
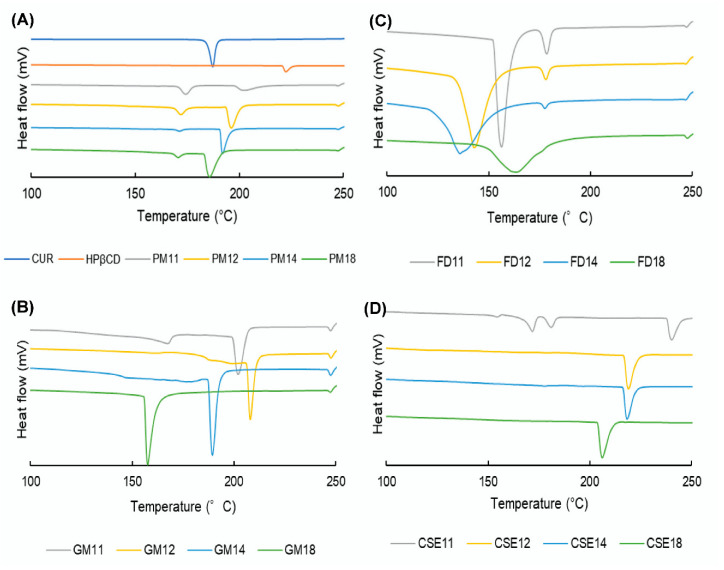
Differential scanning calorimetry curves of: (**A**) curcumin (CUR), hydroxypropyl β cyclodextrin (HPβCD), and physical mixtures (PMs); (**B**) ground mixtures (GMs); (**C**) freeze-drying mixtures (FDs); and (**D**) common solvent evaporation mixtures (CSEs).

**Figure 7 pharmacy-08-00203-f007:**
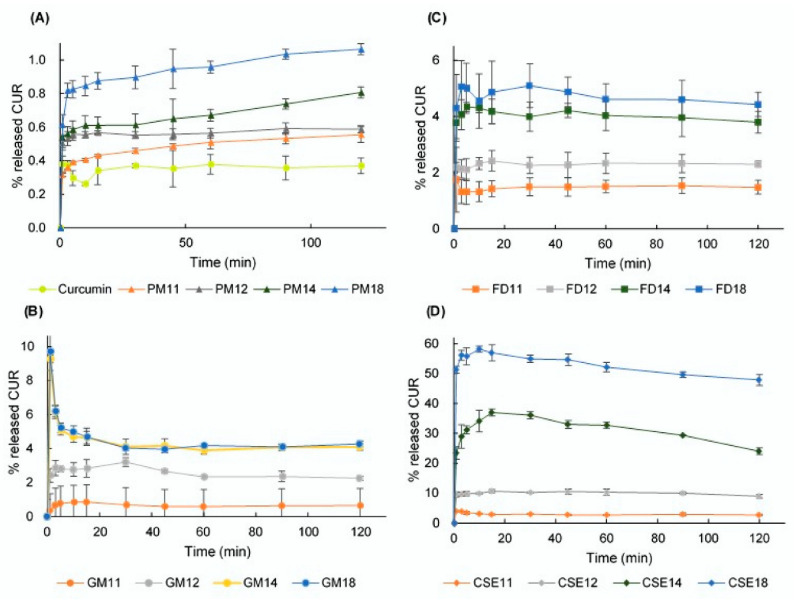
Dissolution profiles of: (**A**) curcumin (CUR), hydroxypropyl β cyclodextrin (HPβCD), and physical mixtures (PMs); (**B**) ground mixtures (GMs); (**C**) freeze-drying mixtures (FDs); and (**D**) common solvent evaporation mixtures (CSEs).

**Table 1 pharmacy-08-00203-t001:** Solubility and dissolution efficiency (DE) of curcumin (CUR) crystals and various curcumin: hydroxypropyl β cyclodextrin solid dispersions prepared using grinding, freeze-drying, and common solvent evaporation methods (*n* = 3).

Sample	CUR:HPβCD Ratio (mol:mol)	Solubility (µg/mL)	DE (%)
CUR	-	0.03 ± 0.01	0.36 ± 0.02
GM11	1:1	0.89 ± 0.17	0.66 ± 0.12
GM12	1:2	1.25 ± 0.25	2.55 ± 0.13
GM14	1:4	2.55 ± 0.24	4.17 ± 0.26
GM18	1:8	6.61 ± 0.15	4.31 ± 0.10
FD11	1:1	0.27 ± 0.02	1.48 ± 0.27
FD12	1:2	0.54 ± 0.01	2.30 ± 0.31
FD14	1:4	1.61 ± 0.13	4.02 ± 0.46
FD18	1:8	5.23 ± 0.18	4.69 ± 0.60
CSE11	1:1	2.64 ± 0.11	2.82 ± 0.18
CSE12	1:2	3.08 ± 0.19	10.00 ± 0.45
CSE14	1:4	4.70 ± 0.29	31.27 ± 0.31
CSE18	1:8	14.16 ± 1.27	52.21 ± 0.75

CUR, Curcumin; HPβCD, Hydroxypropyl β cyclodextrin; GM, Ground mixture; FD, Freeze-drying; CSE, Common solvent evaporation; DE, Dissolution efficiency.

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
