# Peer review of "Enhancing the Solubility of Curcumin Using a Solid Dispersion System with Hydroxypropyl-β-Cyclodextrin Prepared by Grinding, Freeze-Drying, and Common Solvent Evaporation Methods"

_pharmacy, 2020, doi:10.3390/pharmacy8040203_

Round 1

Reviewer 1 Report

The authors report solubilities of curcumin enhanced by solid dispersions systems prepared with hydroxypropyl-beta-cyclodextrin. This topic with curcumin already appears elsewhere. However, the novelty of this study is considered to be comparing preparation methods such as grinding, freeze-drying and solvent evaporation method. Thus, the paper is worthy of publication following minor revision to address the issues mentioned in below.

  1. The authors need to mention the novelty of the current study by comparing published papers that developed the solid dispersion systems of curcumin with hydroxypropyl-beta-cyclodextrin. I could find one paper.
  2. Please add previous approaches to enhance the solubility and dissolution of curcumin by pharmaceutical techniques, along with their advantages and disadvantages.
  3. Curcumin is poorly soluble and thus, visible spectrometry is not suitable due to low levels of curcumin.
  4. The authors need to mention target solubility of curcumin effective for therapeutic outcome with intended route of administration. What is expected dose of curcumin?
  5. Discussion section needs to be improved with citing more papers to scientifically analyze the results obtained.

Author Response

Thank you very much for reviewer’s raising important issues and providing us helpful suggestions. We have performed additional experiments and extensively revised our manuscript in the light of reviewer’s comments.

Response to Reviewer #1

The authors need to mention the novelty of the current study by comparing published papers that developed the solid dispersion systems of curcumin (CUR) with hydroxypropyl-beta-cyclodextrin (HPβCD). I could find one paper.

>Response:

Thank you for your advice. We edited that the novelty of our study is to firstly evaluate the solid dispersion (SD) of CUR and HPβCD instead of conventional inclusion complex of these two components (found papers are 36-38 in Reference section). In this SD system, the inclusion complexes (ICs) will be dispersed in the matrix of hydrophilic CD as a solid dispersion system which further increases drug solubility than the inclusion complex.

Please add previous approaches to enhance the solubility and dissolution of curcumin by pharmaceutical techniques, along with their advantages and disadvantages.

>Response:

Curcumin is poorly soluble and thus, visible spectrometry is not suitable due to low levels of curcumin.

The authors need to mention target solubility of curcumin effective for therapeutic outcome with intended route of administration. What is expected dose of curcumin?

Thank you very much for kind advice. CUR is selected as a model drug due to its very low solubility. If CUR solubility can be increased by forming ICs and non-ICs with HPβCD, other drugs will be easily increased their aqueous solubility by the same way. Because of using as a model, we do not mention therapeutic effects, route of administration, and expected dose of CUR.

Discussion section needs to be improved with citing more papers to scientifically analyze the results obtained.

Thank you for your advice. We cited some papers in interpreting results of phase solubility analysis, FTIR, and DSC study only. Concerning PXRD data, we just used some basic knowledge of crystalline and amorphous phase, so we did not cite any paper. Referring dissolution study, the solid dispersion of CUR and HPβCD was firstly evaluated and the dissolution profiles were transcribed into dissolution efficiency as a comparative data. There is no associated reference.

Reviewer 2 Report

 First, my impression is that the article is written in a way that makes it very hard to follow. It is extremely difficult to be understood for a person without a solid background in solid dispersions and cyclodextrins chemistry, therefore makes it inappropriate for a wide readership. Second, throughout the text, the actual experimental findings are mixed with postulated phenomena (among others, e.g. page 5: experimentally-derived phase solubility analysis vs postulated, without solid experimental evidence like NMR studies, formation of 1:1 and 1:2 complexes - FTIR studies are not enough). Also, other chemical aspects of the paper are not elaborated well enough (e.g., at pH 10 - line 267 - it is hard to talk about an enol form of curcumin, as it is already deprotonated to an extent to an enolate form). Third, the conclusions of the paper seem to combine (probably unintentionally) the actual findings (like the optimum molar ratio) with the concepts or observations already well-established in the literature (like the shifts in IR spectra upon complexation, or reduction in the drug crystallinity causing lower shifts of endothermic peaks). Fourth, in my opinion, the use of an "experimental design" towards such a brief study (12 combinations only) seem like a plan to increase the scientific significance of the study, rather than to be an actually useful approach to the scientific problem. I do appreciate the experimental work behind the study but feel that its design, significance and quality of presentation (editorial mistakes: "Error! Reference source not found."; low quality of some figures) are poor, and the research is rather routine. I recommend the Authors to re-write the manuscript (considering shortening it to include the actual findings), considerably re-working the discussion and conclusions.

Author Response

Thank you very much for reviewer’s raising important issues and providing us helpful suggestions. We have performed additional experiments and extensively revised our manuscript in the light of reviewer’s comments.

Response to Reviewer #2

First, my impression is that the article is written in a way that makes it very hard to follow. It is extremely difficult to be understood for a person without a solid background in solid dispersions and cyclodextrins chemistry, therefore makes it inappropriate for a wide readership.

Thank you for your advice. We added some knowledge about cyclodextrins: history, structure, definition of inclusion complex, their derivatives, and their mechanism in solubility enhancement. I also provided the notion of solid dispersions and preparation methods. We hope this can provide a basic information for readers to follow our manuscript.

Second, throughout the text, the actual experimental findings are mixed with postulated phenomena (among others, e.g. page 5: experimentally-derived phase solubility analysis vs postulated, without solid experimental evidence like NMR studies, formation of 1:1 and 1:2 complexes - FTIR studies are not enough). Also, other chemical aspects of the paper are not elaborated well enough (e.g., at pH 10 - line 267 - it is hard to talk about an enol form of curcumin, as it is already deprotonated to an extent to an enolate form).

Third, the conclusions of the paper seem to combine (probably unintentionally) the actual findings (like the optimum molar ratio) with the concepts or observations already well-established in the literature (like the shifts in IR spectra upon complexation, or reduction in the drug crystallinity causing lower shifts of endothermic peaks).

Thank you for your advice. I edited the conclusion according to the obtained results and the results were previously edited according to reviewers’ comments. Our conclusion was edited as following:

In this study, the SDs of CUR and HPβCD were prepared using the grinding, FD, and CSE methods and were physico-chemically characterized using solubility, PXRD, FTIR, DSC, and dissolution studies. The second or higher order complex of CUR-HPβCD indicated the co-existence of ICs and non-ICs known as the SD system. As comparing the soluble drug amount with CUR crystals, the solubility of SDs was enhanced by up to 299-, 180-, and 489-fold corresponding to the grinding, FD, and CSE preparation methods, respectively. The total transformation into amorphous phase of CUR was observed in ground SDs and in CSE12, CSE14, and CSE18. The drug was well dispersed within HPβCD in ground and CSE SDs whereas the dispersed behavior of FDs was similar to that of PMs. In SDs, the melting temperature of CUR was in an increased order of CUR in 1:2 ICs, CUR in 1:1 ICs, and CUR crystals. The dissolution rate of CUR was positively improved as the amount of HPβCD in SDs increased. The SD system consisting of CUR and HPβCD significantly increased the drug solubility compared to ICs.

Fourth, in my opinion, the use of an "experimental design" towards such a brief study (12 combinations only) seem like a plan to increase the scientific significance of the study, rather than to be an actually useful approach to the scientific problem. I do appreciate the experimental work behind the study but feel that its design, significance and quality of presentation (editorial mistakes: "Error! Reference source not found."; low quality of some figures) are poor, and the research is rather routine. I recommend the Authors to re-write the manuscript (considering shortening it to include the actual findings), considerably re-working the discussion and conclusions.

Thank you for your advice. That is true that the experimental design is used towards a brief study (12 runs only) and we used it to increase the scientific significance of the study. However, we used it with an aim of considering interactions between factors (molar ratio and preparation method), not to designing the experiment. Unfortunately, the analyzed data were similar to the recorded dissolution profiles. Therefore, it is possible not to use the design of experiment in this manuscript.

Reviewer 3 Report

Overall manuscript is well written. Below are few recommendations

  1. The figure are not reference in the paper. Line 165 and line 166
  2. There is line error reference not found on line 55. Similar error on line 157 

Author Response

Thank you very much for reviewer’s raising important issues and providing us helpful suggestions. We have performed additional experiments and extensively revised our manuscript in the light of reviewer’s comments.

Response to Reviewer #3

  1. The figure is not reference in the paper. Line 165 and line 166
  2. There is line error reference not found on line 55. Similar error on line 157 

Thank you for your advice. We checked all editorial errors concerning figures (tables) and reference by having created captions of figures and tables and used cross reference in each referenced paper, in every cited figures and tables.

Round 2

Reviewer 2 Report

Thank you for the opportunity to review the corrected version of the manuscript. I do appreciate the authors' efforts to improve it. The introduction has been extended accordingly, the results and discussion section has been corrected in a very skillful manner. The quality of presentation as well as scientific soundness can now be assessed as high. Owing to the changes made, the manuscript is now of much higher quality and I am delighted to state that it can now be accepted in its present form.